# Epidemic Spread of SARS-CoV-2 Lineage B.1.1.7 in Brazil

**DOI:** 10.3390/v13060984

**Published:** 2021-05-26

**Authors:** Filipe R. R. Moreira, Diego M. Bonfim, Danielle A. G. Zauli, Joice P. Silva, Aline B. Lima, Frederico S. V. Malta, Alessandro C. S. Ferreira, Victor C. Pardini, Wagner C. S. Magalhães, Daniel C. Queiroz, Rafael M. Souza, Victor E. V. Geddes, Walyson C. Costa, Rennan G. Moreira, Nuno R. Faria, Carolina M. Voloch, Renan P. Souza, Renato S. Aguiar

**Affiliations:** 1Departamento de Genética, Instituto de Biologia, Universidade Federal do Rio de Janeiro, Rio de Janeiro 21941-971, Brazil; filipe.moreira@biologia.ufrj.br (F.R.R.M.); carolvoloch@gmail.com (C.M.V.); 2Laboratório de Biologia Integrativa, Departamento de Genética, Ecologia e Evolução, Instituto de Ciências Biológicas, Universidade Federal de Minas Gerais, Belo Horizonte 31270-901, Brazil; t.menezes.diego@gmail.com (D.M.B.); wcsmagalhaes@gmail.com (W.C.S.M.); dcqbioufmg@gmail.com (D.C.Q.); raf.marques.20@gmail.com (R.M.S.); victor.geddes@gmail.com (V.E.V.G.); costa_52@hotmail.com (W.C.C.); renanpedra@gmail.com (R.P.S.); 3Instituto Hermes Pardini, Belo Horizonte 31270-901, Brazil; danielle.zauli@grupopardini.com.br (D.A.G.Z.); joice.silva@grupopardini.com.br (J.P.S.); alinebrito.lima@grupopardini.com.br (A.B.L.); fredericosvm@gmail.com (F.S.V.M.); alessandro.ferreira@grupopardini.com.br (A.C.S.F.); Victor.pardini@grupopardini.com.br (V.C.P.); 4Centro de Laboratórios Multiusuários, Instituto de Ciências Biológicas, Universidade Federal de Minas Gerais, Belo Horizonte 31270-901, Brazil; rennangm@gmail.com; 5Department of Infectious Disease Epidemiology, Imperial College London, London W2 1PG, UK; nfaria@ic.ac.uk; 6Instituto de Medicina Tropical, Faculdade de Medicina da Universidade de São Paulo, São Paulo 21941-971, Brazil; 7Department of Zoology, University of Oxford, Oxford OX1 4BH, UK; 8Instituto D’Or de Pesquisa e Ensino (IDOR), Rio de Janeiro 22281-100, Brazil

## 1. Introduction

The emergence of diverse lineages harboring mutations with functional significance and potentially enhanced transmissibility imposes an increased difficulty on the containment of the SARS-CoV-2 pandemic [1,2,3,4,5,6]. In Brazil, six such lineages cocirculate, one originally from the UK (B.1.1.7) [1], one original from South Africa (B.1.351) [2], and four that emerged within different regions of the country, P.1 (Manaus) [3], P.2 (Rio de Janeiro) [4], N.9 (São Paulo) [5], and N.10 (Maranhão) [6]. While reports on the spread of some of these lineages to other Brazilian regions exist, e.g., [7], a single report on two cases of lineage B.1.1.7 in São Paulo has been published [8], and the extent of its geographic spread is currently unknown. Therefore, we conducted a genomic epidemiology study focused on characterizing the dissemination of this lineage in a national context.

## 2. Materials and Methods

Samples were obtained from the Hermes Pardini Institute (HP), a large Brazilian diagnostic company that performs on average 240,000 COVID-19 tests per month across all Brazilian states. Among other mutations, B.1.1.7 carries the Spike 69/70 deletion leading to the S gene target failure (SGTF) [9] reported in the Thermo Fisher’s COVID-19 assay, used by HP since May 2020. Thus, we retrospectively filtered our dataset for positive samples presenting N gene amplification (Cycle threshold < 30) and SGTF. Among 294,560 samples screened between October 2020 and January 2021, we obtained 25 that met our criteria, collected between 4 January and 24 January 2021 in 8 out of 27 states scattered across four out of the five Brazilian geopolitical regions: northeast (Bahia, Sergipe), central-west (Mato Grosso), southeast (Espírito Santo, Minas Gerais, Rio de Janeiro, and São Paulo) and south (Paraná). Amplified fragments spanning the whole genome of SARS-CoV-2 and DNA libraries were prepared using QIAseq SARS-CoV-2 Primer Panel and QIAseq FX DNA Library Kit, respectively. Sequencing was performed in the Illumina MiSeq instrument. Consensus genome sequences were obtained with a custom pipeline (available on https://github.com/filiperomero2/SC2_lineage_B.1.1.7_in_Brazil, accessed on 24 May 2021) (Appendix A).

Sequences were classified using the PANGOLIN software (https://github.com/cov-lineages/pangolin, accessed on 24 May 2021). To contextualize our findings, a dataset of B.1.1.7 sequences was assembled from data available on GISAID, containing one international sequence per week per country from the discovery of this lineage until 18 February 2021. All 22 previously described Brazilian B.1.1.7 sequences (1 from Goiás, 2 from Distrito Federal, and 19 from São Paulo), and the new genomes generated here were also included. A maximum-likelihood tree was inferred on IQ-tree 2 [10] under the GTR+F+I model, suggested by ModelFinder [11]. Additionally, a dated phylogenetic tree was inferred using TreeTime [12], under a fixed evolutionary rate, as previously estimated [13].

## 3. Results and Discussion

All sequenced samples had genome coverage above 75% (mean 10x coverage: 97.3%, range: 79.8–99.8%; mean 100x coverage: 88%, range: 5–98.8%) and PANGOLIN classified them as lineage B.1.1.7, consistent with the sampling strategy employed. The maximum-likelihood phylogeny supports the occurrence of 13 Brazilian clades containing between two and five sequences from diverse states, suggesting multiple introductions occurred in the country, leading to local transmission chains (Figure 1). While some clades were related to specific states, such as São Paulo or Minas Gerais, others contained sequences from up to three states, revealing the mixing of virus strains across different states driven by human mobility. Eight unique sequences grouped separately with international samples, suggesting at least 21 introductions occurred. This is likely an underestimate, and more extensive phylogeographic analyses remain to be conducted with a higher number of samples and balanced datasets in the future.

It is worth noticing that Brazilian sequences clustered with sequences from several countries from Europe, Asia, and Africa, reinforcing the importance of coordinated global genomic surveillance efforts, and highlighting the need to increase numbers of SARS-CoV-2 sequences from Brazil. The date of emergence of the Brazilian clades was estimated between early December 2020 and early January 2021 (oldest clade: 8 December 2020, 95% CI: 3–9 December 2020; youngest clade: 12 January 2021, 95% CI: 22 December 2020–12 January 2021), consistent with the first report of B.1.1.7 in São Paulo state [5]. Notwithstanding, the first detection of SGTF in HP was on 16 October 2020 in São José do Rio Preto, São Paulo state, suggesting that lineage B.1.1.7 could be circulating even before the dates herein estimated. Further sequencing should confirm this conjecture.

Our results describe the introduction and spread of lineage B.1.1.7 in Brazil. This lineage has now been identified in 10 different states throughout most regions of the country. Considering an estimate of reproduction number obtained in the UK (43–90% higher than previous variants) [14], lineage B.1.1.7 is expected to increase in frequency and lead to a higher number of cases in Brazil. While its fitness advantage in locations with cocirculation with other variants of concern or interest is not entirely clear, future assessments should focus on characterizing the transmissibility of different cocirculating variants across distinct epidemiological backgrounds.

## Figures and Tables

**Figure 1 viruses-13-00984-f001:**
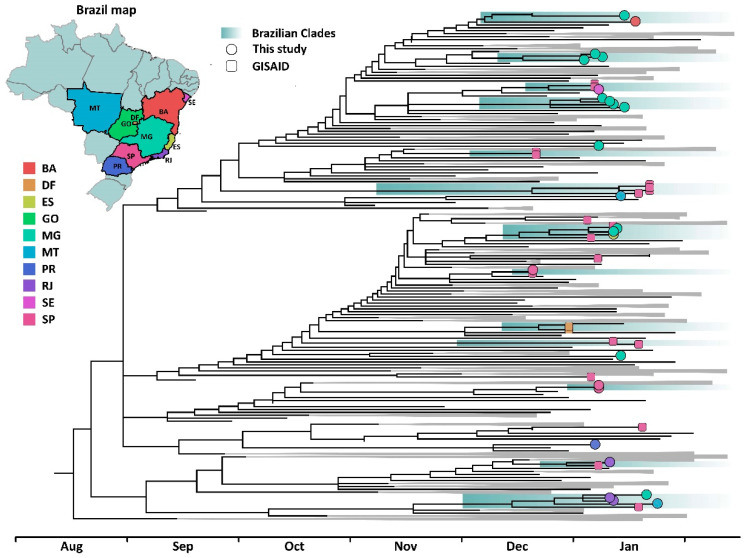
Time scaled phylogenetic tree inferred from a dataset comprehending 406 publicly available B.1.1.7 sequences and the 25 new genomes characterized in this study. Brazilian sequences are color coded according to original federal states and tip shapes mark new and previously described genome sequences. Brazilian clades are shaded in green. The tree supports that multiple introductions occurred in different regions of the country between early December 2020 and early January 2021. While some introductions are related to single sequences, others are linked with the emergence of clades, emphasizing the occurrence of local transmission in the country. Trees inferred in this study are available on https://github.com/filiperomero2/SC2_lineage_B.1.1.7_in_Brazil (accessed on 24 May 2021).

## Data Availability

All generated genome sequences have been deposited on GISAID (IDs: EPI_ISL_1133255 to EPI_ISL_1133279).

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
