# Peer review of "Epidemic Spread of SARS-CoV-2 Lineage B.1.1.7 in Brazil"

_viruses, 2021, doi:10.3390/v13060984_

Round 1

Reviewer 1 Report

Moreira et al., in this study focused on characterizing the dissemination of the B.1.17 lineage in Brazil.

Major concern:

  1. Could the authors explain the criteria they used to select reference sequences?
  2. Author should make available all dataset used in they analysis. Please include them a in free repository
  3. A codon model sounds a bit too complex for this virus. Was the model returned by ModelFinder?
  4. I suggest to deep edit all the references used. Sounds like a auto-citation. Would be nice recognise the work made by other colleagues involved in the SARS-CoV-2 response in Brazil and worldwide. 
  5. Could the authors explain how the were able to estimate the HPD interval by using tree time?

Try to understand the spread of VOCs in Brazil appear to extremely challenging. After those edits I would suggest to accept the present paper for publication in Viruses journal. 

Author Response

Response to reviewer 1 comments

Dear reviewer,

            Thanks for the insightful comments provided during the revision process. Please find below responses to each identified point.

Best Regards,

Renato S Aguiar.

1 - Could the authors explain the criteria they used to select reference sequences?

The dataset used for phylogenetic analysis has been assembled from genome sequences available on GISAID. We downloaded all sequences classified as B.1.1.7 by the GISAID EpiCoV team and composed a reduced representative dataset, comprehending all Brazilian sequences and one international sequence per country per epidemiological week, until the date the analysis was performed. The sampling period corresponds from the first reports of lineage B.1.1.7 to February 18, 2021. To this dataset, we added the novel genomes characterized by our group and conducted all analysis.

2 - Author should make available all dataset used in they analysis. Please include them a in free repository

            We thank the reviewer for the suggestion. However, we are afraid that publicly releasing GISAID EpiCoV data, especially data produced by other groups, might violate this databank policy, possibly implicating in banishment. This regulation can  be directly observed on GISAID frequently asked questions page (https://www.gisaid.org/help/faq/), under the question Do I need permission to use GISAID data when submitting an analysis for publication?

The sequences generated by our group are already available in the GISAID EpiCoV data bank, which is a free repository and accessible to all scientific communities.

3 - A codon model sounds a bit too complex for this virus. Was the model returned by ModelFinder?

            Yes, IQ-Tree’s ModelFinder suggested the nucleotide substitution model GTR+F+I as having the best fit for our alignment. This information has been added to our manuscript to make it clearer to readers. Thanks again for the comment.

4 - I suggest to deep edit all the references used. Sounds like a auto-citation. Would be nice recognise the work made by other colleagues involved in the SARS-CoV-2 response in Brazil and worldwide.

            We thank the reviewer for the suggestion. Following this recommendation, we edited our references to properly cite studies contributed by other groups in Brazil. More specifically, the important studies describing the emergence of N.9 and N.10 variants in the country.

5 - Could the authors explain how the were able to estimate the HPD interval by using tree time?

            We thank the opportunity to provide clarification. TreeTime has a built-in method to calculate confidence intervals for diverge time estimates, calculating the marginal probability distributions of dates of internal nodes. One needs to pass the ‘confidence’ argument, as this option is not used by default. This method documentation is available at https://treetime.readthedocs.io/en/latest/tutorials/timetree.html#confidence-intervals.

Reviewer 2 Report

The study of Moreira et al. entitled “Epidemic spread of SARS-CoV-2 lineage B.1.1.7 in Brazil” the authors conducted a genomic epidemiology study based on SARS-CoV-2 samples presenting undetectable S-gene target collected across Brazilian regions. Phylogenetic analysis reveals all samples belong to lineage B.1.1.7 and suggests the latter was introduced multiple times in the country, leading to transmission chains that comprehend a wider area than previously reported. The theme is important and the authors describe the introduction and spread of lineage B.1.1.7 in Brazil. For these reasons, the manuscript should be accepted after of minor revisions.

Minor revisions:

1) Page 1, line 25: Change to “undetectable S-gene target”

2) Page 1, line 33: Update number of lineages co-circulate in Brazil.

3) Due to the delay in the review, the authors must update the numerical data of the manuscript. For example, page 2, lines 48 to 52, the authors may have processed more samples and increased the number of states in which B.1.1.7 sequences were detected. Also adding the new information in the supplementary material.

Author Response

Response to reviewer 2 comments

Dear reviewer,

            Thanks for the insightful comments provided during the revision process. Please find below responses to each identified point.

Best Regards,

Renato S Aguiar.

1) Page 1, line 25: Change to “undetectable S-gene target”

            We thank the reviewer for the suggestion. The text has been properly changed.

2) Page 1, line 33: Update number of lineages co-circulate in Brazil.

            We thank the reviewer for the suggestion. The number of co-circulating lineages has been updated.

3) Due to the delay in the review, the authors must update the numerical data of the manuscript. For example, page 2, lines 48 to 52, the authors may have processed more samples and increased the number of states in which B.1.1.7 sequences were detected. Also adding the new information in the supplementary material.

            Unfortunately, our group has not expanded the sampling of genome sequences from samples explicitly presenting SGTF. Therefore, the number of sequences included in the scope of this study remains the same. Despite the review delay, the detection of B.1.1.7 sequences in several Brazilian regions is still a matter of concern, since this variant has been associated with increasing rates of severe cases and deaths in the UK and has to be monitored in our country.

Round 2

Reviewer 1 Report

The revised version have been much improved.